# Public Awareness and Behaviour in Great Britain in the Context of Sunlight Exposure and Vitamin D: Results from the First Large-Scale and Representative Survey

**DOI:** 10.3390/ijerph17186924

**Published:** 2020-09-22

**Authors:** Kevin Burchell, Lesley E. Rhodes, Ann R. Webb

**Affiliations:** 1Department of Earth and Environmental Sciences, Faculty of Science and Engineering, University of Manchester, Manchester M13 9PL, UK; kevinwburchell@gmail.com; 2Faculty of Medicine Biology and Health, Division of Musculoskeletal and Dermatological Sciences, School of Biological Sciences, University of Manchester, Manchester M13 9PL, UK; lesley.e.rhodes@manchester.ac.uk; 3Photobiology Unit, Dermatology Research Centre, Salford Royal NHS Foundation Trust, Manchester Academic Health Science Centre, Manchester M6 8HD, UK

**Keywords:** sunlight exposure, vitamin D, public health communication, public knowledge, public awareness, public behaviour, Great Britain

## Abstract

In recent years, UK public health messages about the risks of sunlight exposure (skin cancer) have been increasingly balanced by messages about its benefits (vitamin D production). Currently, data about the effects of this shift on public knowledge, awareness, and behaviour are scant. Thus, the objective of this paper is to report the findings of the first large-scale and representative survey of the awareness, knowledge, and behaviour of adults in Great Britain (England, Scotland, and Wales) (*n* = 2024) with respect to sunlight exposure, vitamin D, and sunburn and skin cancer. The findings suggest that the public in Great Britain is much more aware of public promotion of the risks of sunlight exposure than its benefits. That said, knowledge about sunlight exposure and vitamin D is fairly strong, though not with respect to the detail of the ‘little and often’ approach. However, the survey also suggests that levels of sunlight exposure among the public are often excessive. The survey indicates that knowledge and behaviour are both less satisfactory among men and people in lower socio-economic groups. The paper concludes with recommendations for public health communications and for research in this area.

## 1. Introduction

Vitamin D is an important and unusual nutrient. It is important throughout the human life course because it helps regulate the amount of calcium and phosphate in the body, which in turn keep bones, teeth, and muscles healthy [1]. A lack of vitamin D can lead to a range of musculoskeletal problems. In children, inadequate levels of vitamin D can cause bone deformities such as rickets. In adults, vitamin D deficiency can lead to bone pain and weakness (osteomalacia and osteoporosis) and poor muscle strength, resulting in potential for fractures and falls [2,3]. Links between vitamin D deficiency and a number of other conditions—including respiratory infections, autoimmune disorders, a range of cancers, and cardiovascular disease—have been observed [4]. However, in its 2016 UK review of vitamin D and health, the Scientific Advisory Committee on Nutrition concluded that these data are currently inconclusive [2].

Vitamin D is an unusual nutrient in two interlinked ways. First, although vitamin D is naturally present in some foods, adequate vitamin D cannot be easily obtained from dietary sources other than oily fish. Instead, the body’s principal source of vitamin D is the skin, following exposure to the ultraviolet radiation (UVR) in sunlight, specifically ultraviolet B (UVB) [3]. UVB is absorbed by 7-dehydrocholesterol in the skin and undergoes transformation to pre-vitamin D_3_, the first step in cutaneous synthesis of vitamin D. After that, a heat isomerisation taking several hours leads to formation of vitamin D_3_. It then enters the bloodstream and undergoes hydroxylation in the liver to form 25-hydroxyvitamin D (25(OH)D). This is the major circulating form of vitamin D; 25(OH)D is further hydroxylated in the kidneys to produce 1,25-dihydroxyvitamin D (1,25(OH)_2_D). This is the active form of vitamin D in the body [5].

Vitamin D is also unusual because its main natural source—sunlight—presents risks in terms of sunburn and skin cancer. Crucially, it is these risks that have, over the past 30 years or more, tended to shape global public health messages from government organisations and charities, with an emphasis on sun protection and sun avoidance. For instance, SunSmart campaigns were introduced in Australia in 1988 and in the UK in 2003 [6]. More recently, as concerns about low vitamin D status have grown, the recommendation of vitamin D supplementation has become commonplace, particularly for vulnerable groups and for people who test as vitamin D deficient [2,7]. The fortification of foods with vitamin D is an approach that aims to increase population levels of vitamin D [8]. However, few countries routinely fortify common foodstuffs. Thus, sunlight exposure is typically also recommended [9], in part because sunlight exposure is freely available in many contexts.

In addition, in the past decade, some government organisations and charities—including the SunSmart campaigns—have introduced a ‘little and often’ approach to sunlight exposure into their public health messaging on vitamin D and sunlight exposure [10,11,12]. This approach attempts to balance the risks and benefits of sunlight exposure and also recognises that once initiated the efficiency of vitamin D synthesis declines with continuing exposure [13]. Information about the general public’s awareness of these developments in public health messaging and the extent to which it might have influenced behaviour is scant [14]. In the UK, the ‘little and often’ message was based in part on research by Rhodes et al. [15]. More recently, Webb et al. [16,17] conducted further analysis of Rhodes et al.’s data [15] and data from other human in vivo studies [18,19,20,21,22]. This analysis has allowed more specific ‘little and often’ guidance: in practice, for people with lighter skin, daily (or almost daily) sunlight exposure of unprotected skin of a sufficient surface area for just 10–15 min during the spring and summer months should provide adequate vitamin D to avoid vitamin D deficiency (25(OH)D < 25 nmol/L) all year round. Although even very low levels of sunlight exposure can be harmful to the skin [23,24], Webb et al.’s research [16,17] indicates this will be a relatively safe level of exposure, balancing the benefits of vitamin D production and the risks of skin cancer. This research also suggests that, for people with darker skin in the UK, a more challenging 25–40 min of exposure under the same conditions is required to avoid deficiency, and vitamin D supplements should be considered during the winter months [17].

Globally, the definition of vitamin D deficiency and the measurement of vitamin D levels are somewhat complex and controversial [25,26]. Thus, while some refer to vitamin D deficiency as a ‘global health problem’ and a pandemic [27], others are more cautious [28] or sceptical [25]. In the UK and many other countries, vitamin D status is measured by a blood test, typically using 25(OH)D as a proxy for vitamin D itself. In the UK, vitamin D deficiency is defined as 25(OH)D levels of less than 25 nanomoles per litre (<25 nmol/L). That said, levels of both <30 nmol/L and <50 nmol/L are also used in some countries and studies (a move which obviously returns higher percentages of population deficiency than the lower threshold). 

A major global literature review of vitamin D deficiency prevalence studies has been carried out by [26]. The review concludes that, vitamin D deficiency is ‘very common in most countries around the world’, particularly in the Middle East, China, Mongolia, and India. Further, the study observes that, ‘Probably less than 50% of the world population has an adequate vitamin D status (defined in the study as serum 25(OH)D > 50nmol/L) at least in winter’ ([26] p. 35).

Focusing on temperate climates, in the UK, the National Diet and Nutrition Survey (NDNS) has tracked levels of vitamin D deficiency in the UK population, from 2008/2009–2016/2017 [29]. The 2016/7 survey suggests that—even in July–September, when solar UVB availability is high—more than 1.5 million UK adults (one in 25 or 4% of those aged 19–64) are vitamin D deficient (25(OH)D < 25 nmol/L). This figure rises to a staggering 11 million (almost one in three or 33%) during January–March, following the winter months when solar UVB availability is at its lowest (PHE, 2019). There is also evidence that vitamin D deficiency (25(OH)D < 25 nmol/L) is increasing in the UK. For instance, the NDNS suggests that, among adults aged 19–64 and several other age groups, the incidence of vitamin D deficiency has increased by an average of one percentage point each year over the nine years of the survey [29]. Similarly, UK National Health Service (NHS) admissions data show that the proportion of admitted patients in whom vitamin D deficiency was diagnosed (primary or secondary diagnosis) increased between 2012–2013 (36,912 cases among a total of 15.1 million admissions, 0.0024%) and 2017–2018 (152,892 cases from 16.6 million, 0.0096%) [30]. This represents an increase in proportion of 277% or almost fourfold. Importantly, it is not clear whether these NHS figures reflect increased prevalence of vitamin D deficiency, increased testing for vitamin D (which has revealed levels of vitamin D deficiency, which are higher than previously thought), or a combination of the two. Whichever explanation is correct, these figures appear to point to a significant and possibly increasing UK challenge in this regard.

A major pan-European study by Cashman et al. offers an overall figure for vitamin D deficiency prevalence of 13.0% (17.7% in October to March and 8.3% in April to September) based on the 30 nmol/L threshold [28]. At the 50 nmol/threshold, the prevalence was 40.4%. Thus, while Cashman et al. are cautious about use of the term ‘pandemic’, they conclude that, ‘Vitamin D deficiency is evident throughout the European population at prevalence rates that are concerning and that require action from a public health perspective’ ([28] p. 1033). In the US, using the 50 nmol/L threshold, an overall prevalence rate of 41.6% has been proposed [31]. In Australia, Daly et al. offer a prevalence rate of 31% using the 50 nmol/L threshold, prompting them to conclude that ‘Vitamin D deficiency is common in Australia’ (p. 26) [32].

These studies and relevant literature reviews consistently show that particular groups are at higher risk of vitamin D deficiency than the general population [3,27]. In recent years, people with darker skin who are living in temperate climates have become recognised as a particular at-risk group. The interplay between melanin levels in the skin and solar UVB availability in different climates is an important contributor to this. On one hand, in darker skin types, the production of vitamin D is inhibited by higher levels of melanin in the skin (which absorb UVR and also protect against sunburn and skin cancer). On the other hand, sunlight availability is greater in tropical and sub-tropical than in temperate climates. Thus, as the numbers of darker skinned people living in temperate climates has increased since the latter half of the twentieth century, this group has emerged as one that is particularly vulnerable to vitamin D deficiency.

People who expose their skin to sunlight infrequently are also at greater risk of vitamin D deficiency. This might include people who are elderly, unwell, living in an institution, living a sedentary or indoors lifestyle, people who keep their skin covered for reasons related to culture (such as wearing a hijab or burka) or health (such as photosensitive skin disorders or concerns about skin cancer), or simply personal preference. Infants and pregnant women are also understood to be more prone to vitamin D deficiency. Finally, since sunlight availability varies geographically, so too does the risk of vitamin D deficiency. Thus, regardless of skin colour, people living in temperate and cold climate zones are likely to be more vulnerable to vitamin D deficiency than those living in tropical or sub-tropical zones, particularly during the winter months. Finally, and crucially, research shows that propensity to vitamin D deficiency is related to individual behaviour. For further information, see the reviews by Public Health England and van Shoor and Lips [3,26].

A number of studies around the world—both quantitative and qualitative—have examined public knowledge and behaviour with respect to the sunlight exposure, vitamin D, and skin cancer nexus. Some of these studies can be regarded as population studies while others focus on particular groups. It is not straightforward to directly compare findings across these studies. This is due to the different climatic and cultural contexts in which they were carried out, as well as the varying study objectives and methods. That said, taken as a corpus, these findings offer cause for concern about the extent to which public knowledge and behaviour are likely to ameliorate public health anxieties about the prevalence of vitamin D deficiency.

With respect to public knowledge and awareness, the studies focus on a variety of issues, such as: the benefits of vitamin D, the sources of vitamin D, and awareness and practice of the ‘little and often’ approach to sunlight exposure. Knowledge relating to these issues is largely found to be inadequate in these studies. This is reflected in the use of terms such as: ‘limited’ in a UK focus group study with diverse groups [33], at ‘low levels’ in an Australian focus group study with people who are vulnerable to vitamin D deficiency [34], ‘inadequate’ in a Vietnam population survey [35], ‘not…sufficient’ in a Queensland, Australia quota sample survey [36], ‘low’ in a UK focus group study with people with skin cancer and in a Hong Kong survey of middle-aged and elderly Chinese women [37,38], and ‘lacking’ in an Australian survey of office workers [39]. In only one study—a semi-structured interview study with adults in Saudi Arabia—is public knowledge described as ‘reasonable’ [40].

Turning to public attitudes and behaviour in the context of sunlight exposure, the literature strongly suggests that these are largely driven by concerns about sunburn and skin cancer. For instance, Bonevski et al. state this and note that—although many of their participants felt that they were getting enough sunlight, according to official guidelines—this was despite their concerns about skin cancer rather than due to a deliberate vitamin D acquisition strategy [34]. For their part, both Ho-Pham and Nguyen and Kung and Lee report a ‘negative’ attitude towards sunlight exposure [35,38] and Kotta et al. specifically cite the extent to which public concerns about skin cancer inform sunlight exposure behaviours [33]. Once again, the study in Saudi Arabia offers a different picture, emphasising the importance of extreme temperatures and cultural factors in restricting sunlight exposure [40]. In addition, a UK convenience survey study reports more positive attitudes to sunlight exposure than some of the other studies [41]. Finally, a study of Scottish adolescents’ skin cancer awareness, sun-related behaviours, and tanning attitudes (though not vitamin D) observed poor sun-related practice and low skin cancer awareness [42]. Not surprisingly, all of these studies conclude that public health communications are required to tackle the challenges that are presented by these findings.

## 2. Materials and Methods

The aim of this study was to better understand the knowledge and behaviour of the public in Great Britain (England, Scotland, and Wales) with respect to sunlight exposure, vitamin D, sunburn and skin cancer, and associated public health communications. The objectives of the research were to gather insight in six more specific areas:Knowledge about vitamin D, for instance: its benefits, its sources, and the exposure needed to produce adequate vitamin D.Current sunlight exposure behaviours.Changes in sunlight exposure behaviours over the past ten years.Current perceptions of the extent to which public health communications with respect to sunlight exposure focus on the risks and the benefits.Changes in perceptions of the extent to which public health communications with respect to sunlight exposure focus on the risks and the benefitsThe extent to which these factors vary depending on skin type or demographic characteristics.

The aim and objectives of the study were addressed via an online questionnaire survey administered to a representative sample of 2024 adults (18+) in Great Britain. The survey was developed by the authors, with support from social researchers at Capita plc and YouGov UK plc (YouGov). The survey contained questions relating to the survey objectives. In addition, the survey contained a question about skin type. This drew upon the well-known Fitzpatrick scale: Type I: Pale white skin, always burns, never tans; Type II: White skin, usually burns, tans minimally; Type III: White skin, sometimes burns, tans moderately; Type IV: Olive or light brown skin, burns minimally, tans deeply; Type V: Medium brown skin; Type VI: Dark brown or black skin [43]. Finally, the survey contained questions relating to key demographic variables: gender, age, social group (as a reflection of the occupation of the respondent), region/country in Great Britain, employment status, marital status, and the number of children in the household.

YouGov has a panel of more than 800,000 individuals in Great Britain who have agreed to take part in Omnibus surveys. Emails are sent to panellists selected at random from the base sample. The e-mail invites them to take part in a survey and provides a generic survey link. Once a panel member clicks on the link they are sent to the survey that they are most required for, according to the sample definition and quotas. The responding sample is weighted to provide a reporting sample that is representative by age, gender, social class, region, and level of education. The profile is derived from census data. With respect to skin type, we compared the distribution of skin types in our data with the distribution of ethnicities in the 2011 UK Census of Population (Office of National Statistics, 2011).

Our questions were some of a larger set of questions put to a panel by YouGov. Panellists in turn are drawn from a much wider set of people who have actively expressed willingness to provide their opinions on a whole range of matters. They were assigned to the questionnaire that included our questions in such a way as to have a panel representative of the adult population of the country. Assigning the questions within the questionnaires, and assigning the panel to a given questionnaire is the role of YouGov, as are the associated GDPR assurances. Given our use only of anonymised data, with no recourse to identifying the panellists nor direct interaction with them, and the fact that our questions were one small section of a much wider panel questionnaire delivered independently, ethical approval was not required.

The data were analysed in SPSS (version 25, IBM Corp, Armonk, NY, USA). The analysis focused on frequency data and identifying key differences between categories within the demographic variables highlighted above. These differences—for instance, between men and women—were identified through a combination of examining graphical representations of the frequency data and use of the chi square test. In the context of the chi square test, test results with a significance of <0.05 (or >95%) were considered further and test results with a significance of <0.0005 were prioritised. Within the context of each demographic variable, consistent patterns across a number of survey items were identified for reporting.

## 3. Results

Frequency and percentage data—broken down by the demographic characteristics of the respondents and across all of the survey questions—are available in Appendix A. It is important to note that direct comparisons between skin type and ethnicity (Appendix A) are not possible due to variations in skin type within some of the ethnic categories. That said, although not identical, the distributions were very similar, leading us to conclude that our data are broadly representative by skin type. The survey was administered by YouGov on 17–18 June 2019. Where percentages in the Figures in this section do not add up to 100, this is due to rounding anomalies.

### 3.1. Awareness of the Promotion of Risks and Benefits

In the first three parts of the Results section, we describe the results for the whole dataset. Thereafter, we examine the key demographic characteristics. It is helpful to frame our comments on public knowledge and behaviour within the context of the findings with respect to public awareness of the promotion of the risks and benefits of sunlight exposure. As indicated in the Introduction, earlier emphasis on the risks of sunlight exposure (e.g., sunburn and skin cancer) is increasingly balanced in public health communications by messages that also highlight benefits (e.g., vitamin D production). Figure 1 and Figure 2 offer clear evidence that public awareness is lagging behind these developments. Figure 1 shows respondents’ perceptions of the promotion of the risks and benefits of sunlight exposure. More specifically, Figure 1 addresses public perceptions of how well or not the risks (sunburn and skin cancer) and the benefits (vitamin D for strong bones and muscles) are promoted. Figure 2 contextualises these findings in terms of respondents’ perceptions of whether the promotion of the risks and benefits has increased, decreased, or stayed the same over the past ten years.

The results are stark. Clearly, as shown in Figure 1, while many members of the public feel that the risks of sunlight exposure are well promoted (71%) a strikingly similar percentage feel that the benefits of sunlight exposure are not well promoted (69%). Further, Figure 2 suggests that more than half (52%) of respondents feel that promotion of the risks of sunlight exposure has increased over the past ten years. Meanwhile, the corresponding figure for the benefits of sunlight exposure is—despite the introduction of promotion of the ‘little and often’ approach’—just a quarter (24%); here, a further half (48%) feel that promotion of the benefits of sunlight exposure has stayed the same over the past ten years. It is also perhaps significant that the ‘Don’t know’ figures are higher for the benefits of sunlight exposure than for the risks, with respect to perceptions of both promotion and changes in promotion.

### 3.2. Knowledge

Figure 3 reports the survey results with respect to knowledge of vitamin D and sunlight issues, including knowledge about the ‘little and often’ approach. To properly interpret Figure 3, please note that, for each item, the top bar indicates the percentage of respondents who answered the question correctly. Overall, the results suggest that knowledge is strong in some respects. For instance, 74% correctly agreed with the statement ‘Vitamin D is needed for strong bones and muscles’, 85% rightly disagreed with the statement, ‘Vitamin D is important for children but not for adults’, and 78% correctly agreed with the statement, ‘Your skin can make vitamin D if it is exposed to sunlight’. With respect to the ‘little and often’ approach to sunlight exposure, 68% correctly noted that daily or almost daily exposure is important. It is important to be somewhat cautious about this figure. This is because advice about sunlight exposure tends to be quite specific in terms of the time of day and the parts of the body that are exposed. It was not possible to capture these nuances in the survey.

As Figure 3 shows, more mixed results were also observed: 42% wrongly agreed with the statement, ‘A balanced diet will give most people enough vitamin D’ (with 39% correctly disagreeing); further, 40% wrongly agreed that, ‘Vitamin D prevents scurvy’ (with 34% rightly disagreeing). Returning to the ‘little and often’ approach, it is notable that fewer than half correctly identified times between 10 to 30 min as ideal lengths of daily exposure. So that they were able to answer this question, survey respondents were first informed that ‘Research shows that daily exposure to sunlight is important for making sufficient levels of vitamin D’. For this reason, it is important to be cautious about this figure. In addition, caution should be applied because different lengths of time are recommended for different skin types and the specific advice given can vary. Thus, to cover a range of skin types, the answers accepted as ‘correct’ were ‘More than 10 min a day, up to 15 min a day’ and ‘More than 15 min a day, up to 30 min a day’.

### 3.3. Behaviour

Turning to behaviour, Figure 4 shows the survey results relating to people’s estimates of their daily sunlight exposure during the summer months (on weekdays and weekends). We have compared the survey results with recent public health advice regarding the ‘little and often’ approach to sunlight exposure. However, this is not straightforward for a number of reasons. First, the advice varies in different fora and varies for different people. Further caution should be applied here due to: the challenges that people might have in encapsulating their varying daily behaviours within a single response, and because the data do not take account of the time of day of the exposure, varying levels of sunlight availability (either geographically or from day-to-day), the parts of the body that are exposed, and the use of sunscreen (all of which can be significant factors in determining appropriate levels of sunlight exposure). Thus, for the purposes of this commentary, we have included ‘Up to 15 min’, ‘More than 15 min, up to 30 min’, and ‘More than 30 min, up to 1 h’ within the range of sunlight exposure behaviours that might be broadly acceptable within the context of a ‘little and often’ approach.

Even bearing these caveats in mind, there is evidence here that many or even most people may not currently be following the ‘little and often’ approach. Indeed, the data in Figure 4 suggest that many people—60% on weekdays and 77% at weekends—might be exceeding the levels of sunlight exposure that are recommended within a ‘little and often’ approach, according to the broad-based parameters that we have used here. This finding is concerning in terms of sunburn and skin cancer prevention. In terms of insufficient daily exposure and the risk of vitamin D deficiency, at least 3% of respondents might fall into this category (both weekdays and weekends). Meanwhile, Figure 4 suggests that just 31% on weekdays and 15% on weekends are—by accident or design—performing daily sunlight exposure behaviour that is within the ‘little and often’ parameters as defined here.

Figure 5 reports the results with respect to the frequency with which people perform two specific sunlight exposure behaviours during the spring and summer months (April to September). Figure 6 contextualises Figure 5 in terms of respondents’ judgements of the extent to which this frequency has increased, decreased, or not changed over the past ten years. There are, perhaps, both encouraging and discouraging messages in these results. For instance, although 19% of respondents say that they have increased the frequency with which they follow the ‘little and often’ approach, 14% say that they do this less often (yielding a net increase of just 5%). Meantime, the 16% who never get regular, short exposures to the sun (Figure 5) should be of concern as at risk of vitamin D deficiency. In addition, healthcare professionals would surely like to see the numbers of people who always/often ensure regular exposure for short periods increase from the current 33%. Similarly, although 24% say that they ‘get as much exposure to the sun as possible’ less often than in the past, 14% say that they do this more now (yielding a net decrease of just 10%). In addition, it is of concern that 22% always/often and 40% sometimes currently get as much sunlight exposure as possible.

### 3.4. Demographic Characteristics

#### 3.4.1. Introduction

As discussed in the Methods section, part of our analysis focused on identifying differences or distinctions between categories—for instance, between men and women—within the context of a number of demographic attributes: gender, age, social group, region/country in Great Britain, employment status, marital status, and the presence of children in the household). In addition, we examined the significance of skin type. As also noted in the Methods, the analysis involved the identification of significant differences and of consistent patterns across a number of survey items for reporting. In terms of public health interventions, this part of our analysis has the potential to facilitate the targeting of messages about sunlight exposure and vitamin D, and we return to this in the discussion.

Among the demographic variables, consistent patterns were most emphatically observed within the context of gender and social group; these variables are examined in detail. Age, employment status, and skin types emerged as less variable, so these are only briefly discussed. The analysis revealed minimal significant differences and certainly no discernible patterns with respect to marital status, the presence of children in the household, and region/country in Great Britain.

#### 3.4.2. Gender

Among the demographic variables, consistent patterns were most emphatically observed within the context of gender. For this reason, for illustrative purposes, all of the results for gender are reported here. As Figure 7 shows, we analysed the survey items relating to knowledge, behaviour, and the promotion of risks and benefits. The survey items are listed on the left in edited form. Alongside each item, a single dot indicates a significant difference at 0.05 (95%) and two dots signify a significant difference at 0.0005 (99.95%). In each case, the top bar indicates the percentage of respondents who gave the correct answer to a knowledge question or an answer that reflects a desired behaviour in the context of the behaviour questions. Figure 7 indicates that women have somewhat better knowledge about sunlight exposure and vitamin D. This is the case with respect to six out of the eight Knowledge items, while one item—relating to the (false) relationship between vitamin D and scurvy—indicates greater knowledge among men. However, Figure 7 also shows that this difference in knowledge does not translate into a consistent difference in behaviour; in this context, it is only the item relating to performance of a ‘little and ‘often’ approach on weekdays where a difference is apparent (again, women’s behaviour appears to be more likely to—deliberately or not—follow a ‘little and often’ approach). Finally, Figure 7 indicates that women and men perceive the promotion of the risks and benefits of sunlight exposure similarly.

#### 3.4.3. Socio-Economic Group

The data allow us to distinguish between higher socio-economic groups (ABC1, those in managerial, administrative, or professional roles) and lower socio-economic groups (C2DE, those in manual/unskilled roles or dependent in some way on the state), based upon the occupation of the main earner in the household. Figure 8 shows only the significant findings relating to socio-economic group. Figure 8 suggests that there is some—albeit, limited—evidence that people in higher socio-economic groups have more knowledge about sunlight exposure and vitamin D than people in lower socio-economic groups. Importantly, these differences are not as sizeable or consistent as those between men and women. Figure 8 also contains very limited evidence of differences in behaviour. Again, people in higher socio-economic groups appear to be a little more likely in some contexts to perform behaviours that more closely align with the ‘little and often’ approach.

#### 3.4.4. Other Variables

As discussed earlier, it is possible to draw only very tentative conclusions within the context of age, employment status, and skin type:Age: There is some evidence that older people are likely to be more knowledgeable about sunlight exposure and vitamin D than younger people. There is also evidence that people aged 55 and over—which, unfortunately, is a very large and varied category—may be more likely to exceed the levels of sunlight exposure that are recommended within the ‘little and often’ approach.Employment status: There is some evidence that people who are working are likely to know more about sunlight exposure and vitamin D than people who are not working (unemployed or retired). The sample is not representative by employment status.Skin type: As discussed earlier, we estimate that the sample is broadly representative by skin type. One implication of this is that the numbers of respondents in the darker skin categories are relatively small (see Appendix A). There is evidence (not consistent across the relevant questions) that people with skin type VI (dark brown or black) may be likely to be more knowledgeable about sunlight exposure and vitamin D than people with lighter skin types. There is also inconsistent evidence that people with skin types I and II (those most likely to burn/least likely to tan) are more likely to conform to the prescriptions of the ‘little and often’ approach to sunlight exposure.

## 4. Discussion

The objective of this paper is to examine—via a representative survey—the awareness, knowledge, and behaviour of adults in Great Britain with respect to sunlight exposure and vitamin D. This is an important issue within the context of high levels of vitamin D deficiency in the UK and elsewhere, with attendant concerns for bone and muscle health and other potential health benefits of vitamin D and sun exposure [4,44]. The paper suggests that UK adults are relatively unaware of the promotion of the benefits of sunlight exposure. The paper also shows that, despite this, knowledge about sunlight exposure and vitamin D is strong in some respects (e.g., the benefits of sunlight exposure/vitamin D and the role of regular sunlight exposure in vitamin D production) though more mixed in others (e.g., the lack of availability of vitamin D from dietary sources). Further, in broad terms, the paper indicates that such knowledge is more likely to be lacking among men (this has been shown in some earlier studies [36,39]) and people in lower socio-economic groups (as well as, with less certainty, people who are not working and younger people). 

More concerningly, the paper shows that—as has been shown across a wide range of issues—knowledge does not straightforwardly translate into positive behaviour in this context. The data presented here suggest that around two-thirds of people are not following—and most of these are exceeding—the parameters of a ‘little and often’ approach to sunlight exposure. In this context, the difference between men and women disappears and the difference between higher and lower socio-economic groups is reduced. Some cause for optimism is embedded in the observation that some people observed that their behaviour was changing positively in this regard. While concerns about sunburn and skin cancer appear to inform behaviour in some of the other countries in which research has been carried out [34,35,38], in Great Britain, awareness of the promotion of the risks of sunlight exposure does not appear to be reflected in behaviour. A positive finding of the study is that it appears that groups who are more at risk of sunburn and skin cancer (people with very light skin) and one of the groups that is most at risk of vitamin D deficiency (people with dark brown or black skin) were somewhat distinctive in terms of knowledge and behaviour; targeted communications in these areas appear to have been successful to some extent.

That said, the findings of this survey appear paradoxical on a number of levels. In clinical contexts, there is evidence to support concerns about increasing levels of vitamin D deficiency, and these are increasingly reflected in public health communications about sunlight exposure. However, the public appears to be—as yet—relatively unaware of these changing communications (especially compared to communications about the risks of sunlight exposure). Despite this, levels of knowledge about sunlight exposure and vitamin D are healthy (though not, perhaps, the detail of the ‘little and often’ approach). Not surprisingly, lack of knowledge about the ‘little and often’ approach is reflected in relatively little adherence to this approach in people’s behaviour. In a further twist, awareness of communications regarding the risks of sunlight exposure appear to be contradicted by widespread behaviour involving higher-than-recommended levels of sunlight exposure. For consistency with other evidence, this would imply that the longer (assumed over-) exposures occur intermittently (e.g., at weekends, holidays) or early/late in the day when the sun is low in the sky and there is little UVB meaning the dose received is nonetheless small.

These findings inform a number of recommendations for research and practice and at the intersection of the two. In response to international interest in vitamin D, comparative international survey research into awareness, knowledge, and behaviour would be of great value. Ideally, this and other research will be able to take account of the role of supplementation, which was outside of the remit of this study. It would be of value to complement quantitative survey work with further in-depth interview and group-based qualitative research; in particular, this could facilitate detailed investigation and disentanglement of some of the apparent paradoxes that this research seems to have uncovered.

In terms of the practice of public health communications in this area, the findings suggest that it is very important to increasingly emphasise the detail and purpose of a ‘little and often’ approach to sunlight exposure. Importantly, the findings suggest that this is important within the contexts of both the risks of sunlight exposure in terms of skin cancer and its benefits in terms of vitamin D production. This study provides evidence that specifically targeting men and people in lower socio-economic groups would be of particular value. However, this is a relatively complex public health message, largely due to the tension between too much and too little sunlight exposure and the varying advice for different groups within the general population (for instance, the distinction between lighter and darker skin types). This raises the importance of social or behavioural studies that combine research and practice—perhaps quasi randomised-controlled trials (RCTs)—in which the behavioural and attitudinal impact of a variety of communication strategies, communication channels, and communication messages could be studied both quantitatively and qualitatively.

## 5. Conclusions

In Great Britain, following some 30 years of public health communications that have emphasised sunlight exposure protection and avoidance, communications now increasingly focus on a ‘little and often’ approach to sunlight exposure. This shift in Great Britain, reflecting a recognition of the value of sunlight exposure for vitamin D production and the—possibly increasing—prevalence of vitamin D deficiency, is mirrored in other countries around the world. Within this context, with the objective of better understanding public awareness, knowledge, and behaviour with respect to these issues, this paper has reported the findings of the first large-scale and representative survey on this topic in Great Britain. Thus, the findings from this survey are likely to be of interest to public health professionals, charities, researchers, and other stakeholders with an interest in sunlight exposure and its risks and benefits. 

The survey suggests that the adult public in Great Britain is much more aware of communications related to the risks of sunlight exposure than its benefits. Somewhat paradoxically, the survey also suggests that public knowledge about sunlight and vitamin D is fairly strong, though not with respect to the detail of the ‘little and often’ approach. Further, and perhaps alarmingly, the survey suggests that many people in Great Britain might be routinely experiencing levels of sunlight exposure that are excessive in terms of sunburn and skin cancer but fail to provide a commensurate vitamin D benefit. The survey suggests that challenges in these regards are likely to be most pronounced among men and people in lower socio-economic groups.

In response to these findings, we have suggested that there is a need for greater emphasis on the ‘little and often’ approach in public health communication by charities and government organisations. Crucially, this could be beneficial in terms of communicating both the risks and the benefits of sunlight exposure. Further, we have suggested that these communications could usefully focus on men and people in lower socio-economic groups. We have also suggested that there is a need for further research in this area. International, comparative research would be of value, as would complementary qualitative research. Finally, given the complexity and importance of the ‘little and often’ message, we have suggested that social and behavioural research should be employed to ‘test’ different messages and communications channels.

## Figures and Tables

**Figure 1 ijerph-17-06924-f001:**
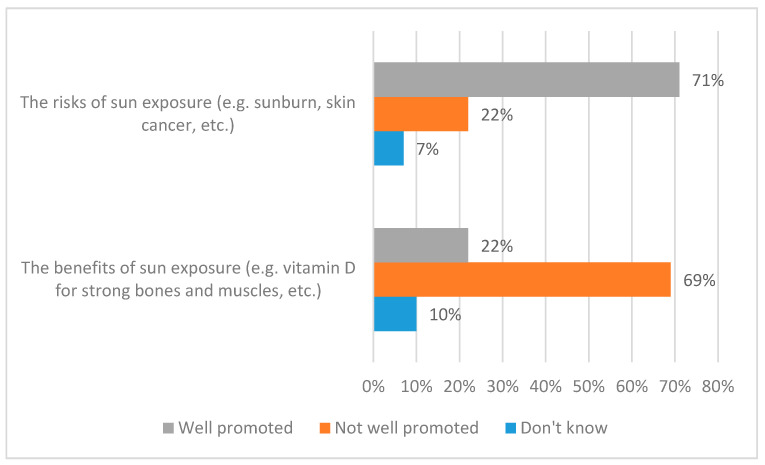
Public awareness of the promotion of the risks and benefits of sunlight exposure.

**Figure 2 ijerph-17-06924-f002:**
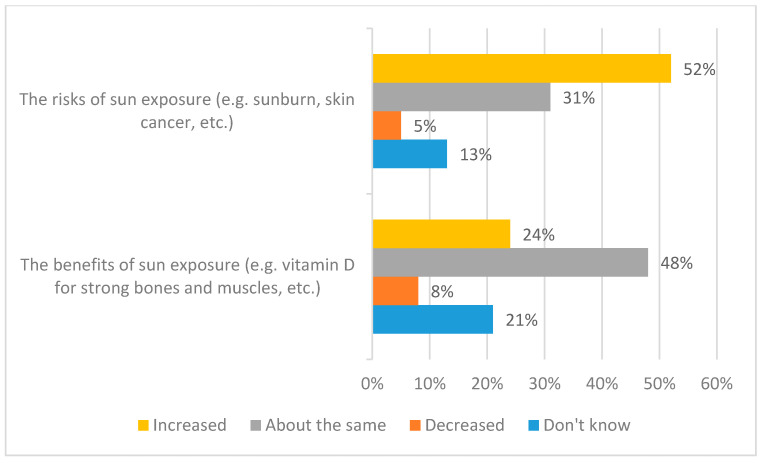
Awareness of change in the promotion of the risks and benefits of sunlight exposure over the past ten years.

**Figure 3 ijerph-17-06924-f003:**
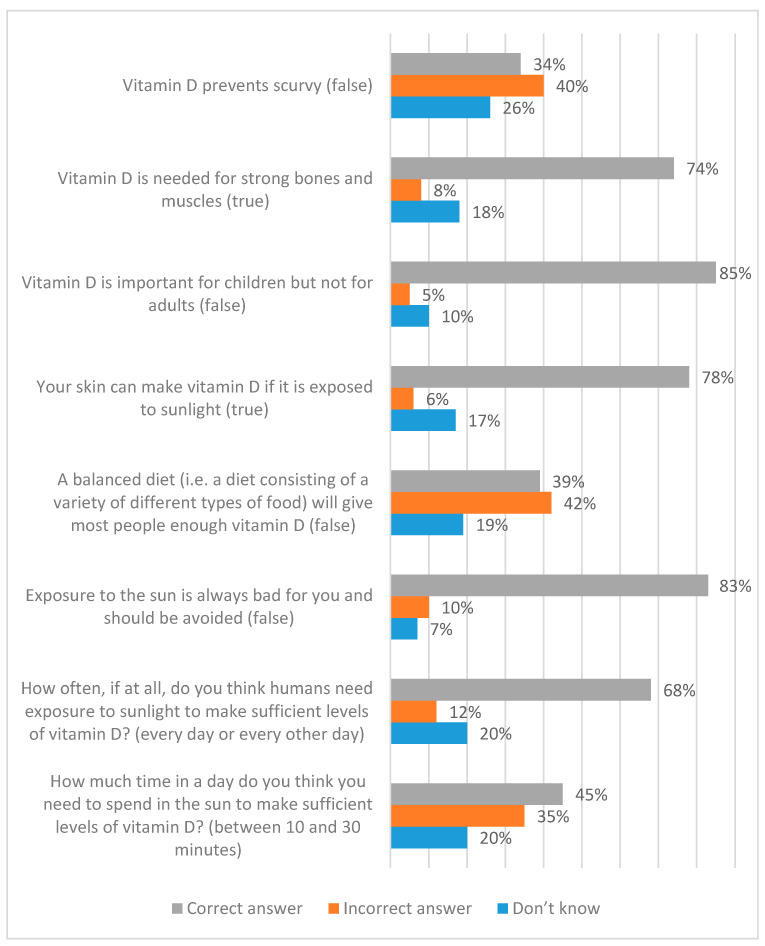
Public knowledge about vitamin D and sunlight exposure.

**Figure 4 ijerph-17-06924-f004:**
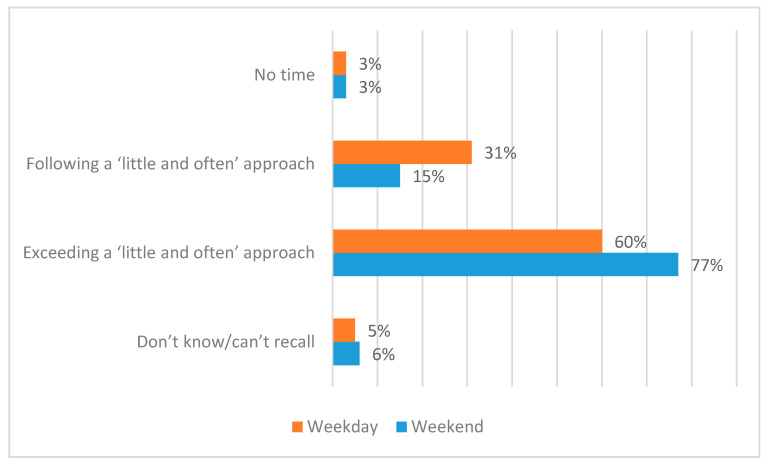
Daily sunlight exposure behaviours.

**Figure 5 ijerph-17-06924-f005:**
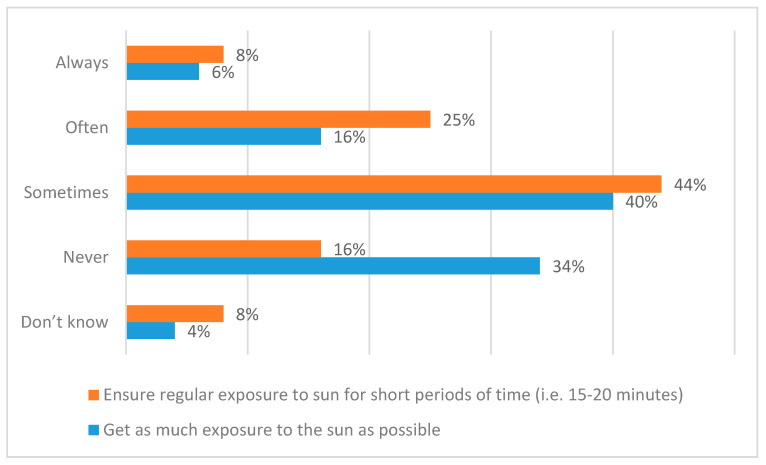
Frequency of specific sunlight exposure behaviours (spring and summer months, April to September).

**Figure 6 ijerph-17-06924-f006:**
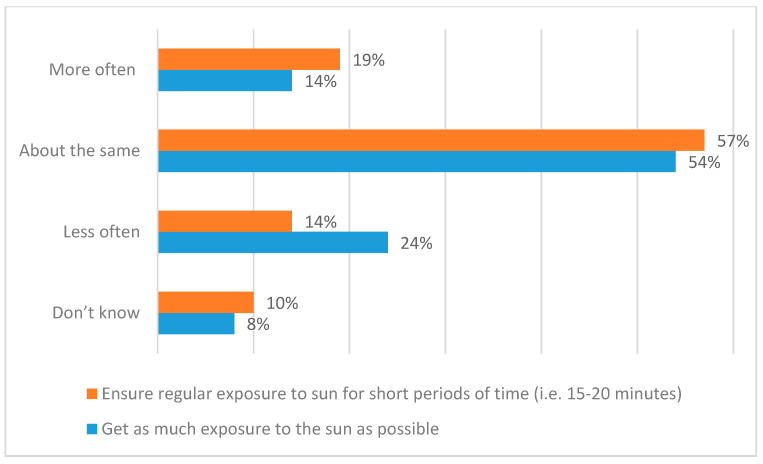
Extent to which these frequencies have increased, decreased, or not changed over the past ten years.

**Figure 7 ijerph-17-06924-f007:**
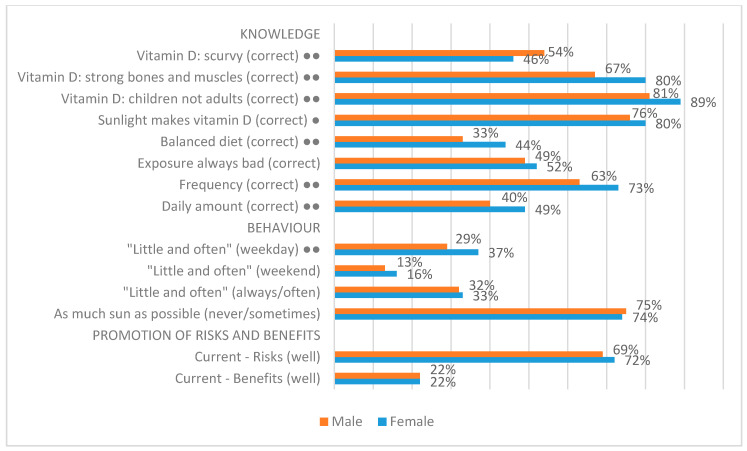
Gender differences in survey responses. A single dot indicates a significant difference in response at 0.05 (95%) and two dots signify a significant difference at 0.0005 (99.95%).

**Figure 8 ijerph-17-06924-f008:**
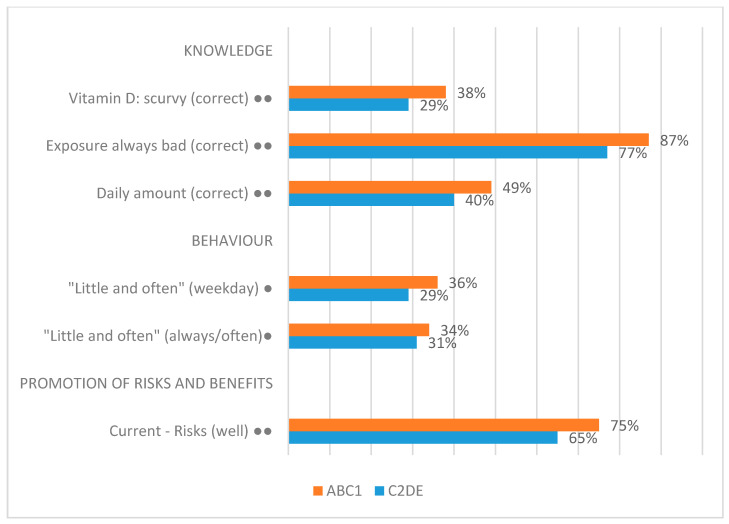
Significant socio-economic group differences in survey responses. A single dot indicates a significant difference in response at 0.05 (95%) and two dots signify a significant difference at 0.0005 (99.95%).

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
