# Peer review of "Public Awareness and Behaviour in Great Britain in the Context of Sunlight Exposure and Vitamin D: Results from the First Large-Scale and Representative Survey"

_ijerph, 2020, doi:10.3390/ijerph17186924_

Round 1

Reviewer 1 Report

The authors are to be congratulated on an excellent, informative piece of work. The introduction sets the scene in a most appropriate manner, covering all important aspects to a complex issue.

The six areas covered by the survey are adequate and address all of the important components that need consideration of a study with the aim to better understand knowledge and behaviour of the public with respect to sun exposure, vitamin D, sunburn and skin cancer. I was pleased to see that FST was included.

I am delighted to see that the authors used an existing panel such as those used in Omnibus surveys. This is excellent information to share to fellow researchers for new ways in which to access participants.

The methods seem appropriate, only I was somewhat disappointed that only SPSS was applied to look at frequencies and key differences. Surely more powerful work with these data would be possible. Perhaps for another paper.

The results are certainly intriguing. Figures 1 and 2 are almost inverse of each other for the risk versus the benefit awareness.

It is a little tricky to see the text in Figure 7. It is very small.

Lines 384-386 are surely surprising regarding responses from deeply-pigmented skin. Could this be skewed by very small numbers in this category according to the Supplementary data?

I would prefer that the Supplementary data be included in the main paper so avoid having to refer between two documents all the time.

I certainly agree that the results so seem paradoxical at times but this has been the case in many other surveys over the years on the topic of sun exposure risks and benefits. As explained in the last p/g of the discussion, finding the right advice is so tricky. I agree that the time has come for more integrated approaches to solving these challenges such as research and practice and quantitative and qualitative - these are good observations.

Author Response

Reviewer 1:

Thank you for your kind comments on our work. The responses to your queries are provided below.

  1. It is a little tricky to see the text in Figure 7. It is very small.

A larger version of the figure was supplied to the journal on submission. The layout of the final paper, to achieve sufficiently large text, will be decided by the journal editors.

  1. Lines 384-386 are surely surprising regarding responses from deeply-pigmented skin. Could this be skewed by very small numbers in this category according to the Supplementary data?

Yes, this is possible, there were limited respondents identifying as dark-skinned. However, we do not find it particularly surprising as those most likely to suffer from vitamin D deficiency are perhaps most likely to learn that a major source is sun exposure. Knowledge, and ability to apply that knowledge (hampered by limited sun in UK), are different.

We have modified the text to say: Skin type: As discussed earlier, we estimate that the sample is broadly representative by skin type. One implication of this is that the numbers of respondents in the darker skin categories are relatively small (see Supplementary Table 2). There is evidence (not consistent across the relevant questions) that people with skin type VI (dark brown or black) may be likely to be more knowledgeable about sunlight exposure and vitamin D than people with lighter skin types. There is also inconsistent evidence that people with skin types I and II (those most likely to burn / least likely to tan) are more likely to conform to the prescriptions of the ‘little and often’ approach to sunlight exposure.

  1. I would prefer that the Supplementary data be included in the main paper so avoid having to refer between two documents all the time.

In response to a request from Reviewer 2, we are now including very comprehensive supplementary data: counts and percentages – broken down by key demographic variables – across all of the survey questionnaires. Therefore, we consider that this data is best expressed as Supplementary material.

Reviewer 2 Report

The manuscript by Burchell and colleagues reports on a survey examining the awareness, knowledge and behaviour of 2024 British (except for Northern Islanders) adults about sun exposure risks and benefits, which include skin cancer and adequate vitamin D levels, respectively.

This is a valuable study that provides an indication of the extent to which public health messages about the advantages and disadvantages of sun exposure are absorbed by the target population, and lead, or not, to the anticipated behavioural changes. It also assesses whether the changes that these messages have experienced in time are perceived and reacted to by the intended targets. Given its characteristics, the survey used only allows for obtaining suggestive information and not for an in depth examination of the current sun exposure-related habits in Britain and of potential intervention strategies, which deserve further analysis in the future.

Comments

  1. The introduction is quite comprehensive, describing in detail the state of vitamin D deficiency across populations in general and in the UK in particular. However, there are other benefits to sun exposure (for example, those discussed by Lucas and Rodney-Harris[1]), likely independent of vitamin D, and although the evidence is not as strong as for vitamin D so far, I believe it would be worth briefly mentioning them.
  2. Introduction, line 111: the sentence should refer to “vitamin D deficiency prevalence...”, the word deficiency is missing.
  3. Introduction, lines 121-122: although I understand what it is meant, it nevertheless sounds a bit odd to say that "people with darker skin… have become recognised as an important group". It could be changed to: as an important at-risk group or something similar.
  4. Introduction, lines 131-134: I would also add preference as a reason for people rarely exposing their skin to the sun. Some people just do not like/enjoy being exposed to the sun.
  5. Introduction, line 140: the reference by van Shoor and Lips is #26 not 27.
  6. Materials and Methods, line 188: would it be possible to include the questionnaire used as supplementary material?
  7. Materials and Methods, line 205: what does it mean “industry accepted data”?
  8. Materials and Methods: was level of education available in this survey? Even though socioeconomic position is correlated with education, this correlation will not be perfect, and assessing knowledge about vitamin D and sunlight exposure risks and benefits by education level as well would be relevant to the study’s objectives. It may strengthen the differences depicted in Figure 8. In addition, as other studies on sun exposure practices also examined educational attainment, it would make findings from this study more comparable with the literature.
  9. Results, line 296: based on Figure 4 it should be 77% at weekends, not 71%.
  10. Results, lines 355-356: please define more explicitly what socioeconomic positions ABC1 and C2DE mean.
  11. Results, other variables: please include these results as supplementary material.
  12. Discussion, line 395: reference is out of format.
  13. Discussion, lines 409-412: please provide some references to the “other countries” affirmation.
  14. Discussion: it would be good to briefly note a) results from other studies that have shown differences between men and women in terms of knowledge and actual sun exposure (this has been seen quite frequently across studies), and b) other surveys carried out in Britain that examined sun-related behaviours and skin cancer awareness, for example that reported by Kyle et al. (2014)[2] in Scottish adolescents.
  15. I think that the answer “get as much exposure to the sun as possible” does not necessarily imply going above the acceptable ranges. Great Britain does not get a lot of sun and maybe people who exposed themselves to the sun whenever it is out are just following public health advise to try and synthesize enough vitamin D and they are not overdoing it. Thus, this answer may or may not represent a risky behaviour. Obviously, if respondents are doing this abroad then it is likely to be risky. Was there any question related to where the individuals were getting their sun exposure? Or any more detail provided in their answers so as to tease out who is exposed excessively and who is doing something comparable to “little and often”?
  16. I suggest the authors include a comment on the 3% of individuals who dedicate no time to sunlight exposure (Figure 4), and on the 16% of respondents who never ensure regular sun exposure (Figure 5), as they may be at risk of vitamin D deficiency and other negative consequences of low or inexistent exposure. A message geared towards these people would also be important.
  17. Figures 1-5: make sure that percentages sum up to 100%.
  18. Figure 8: the bars for current – benefits are missing.
  19. Supplementary Table 2: please substitute Carib for Carrib.

References

  1. Lucas, R.M.; Rodney-Harris, R. Benefits of sun exposure : vitamin D and beyond. In Proceedings of the NIWA UV Workshop; 2018; Vol. 2018.
  2. Kyle, R.G.; MacMillan, I.; Forbat, L.; Neal, R.D.; O’Carroll, R.E.; Haw, S.; Hubbard, G. Scottish adolescents’ sun-related behaviours, tanning attitudes and associations with skin cancer awareness: a cross-sectional study. BMJ Open 2014, 4, e005137.
